# Young People Awareness of Sexually Transmitted Diseases and Contraception: A Portuguese Population-Based Cross-Sectional Study

**DOI:** 10.3390/ijerph192113933

**Published:** 2022-10-26

**Authors:** Carlos Franclim Silva, Inês Silva, Alexandra Rodrigues, Luísa Sá, Daniel Beirão, Paula Rocha, Paulo Santos

**Affiliations:** 1Department of Community Medicine, Information and Health Decision Sciences, Faculty of Medicine, University of Porto, Alameda Hernani Monteiro, 4200-319 Porto, Portugal; 2CINTESIS@RISE, MEDCIDS, Faculty of Medicine, University of Porto, Alameda Hernani Monteiro, 4200-319 Porto, Portugal; 3Unidade de Saúde Familiar São João do Porto, Rua Miguel Bombarda, 234, 4050-377 Porto, Portugal; 4Serviço de Dermatologia do Centro Hospitalar e Universitário do Porto, Largo do Prof. Abel Salazar, 4099-001 Porto, Portugal; 5Center for Research in Higher Education Policies (CIPES), Universities of Aveiro and Porto, Rua 1º Dezembro 399, 4450-227 Matosinhos, Portugal

**Keywords:** adolescent, young adult, sexually transmitted diseases, contraception, health education, social planning, sex education, sexuality

## Abstract

Adolescents and young adults are an important target concerning reducing health-risk behavior adoption, including sexual health. Studying their knowledge concerning sexuality and their main counsellors, can be an important step in targeting an updated health promotion approach. This study characterized adolescents and young adults’ knowledge and attitudes about sexually transmitted diseases (STDs), and contraception, prospecting for their main trusted counseling sources. We conducted a cross-sectional, population-based, self-report survey of 746 individuals aged between 14 and 24 years from Paredes, Portugal. The questionnaire included many dimensions, as demographic characteristics, youth behavior, currently sexually active status, main counselors concerning health topics, awareness, and knowledge about STDs and contraception. Mean age of the participants was 18.3 years, 50.5% of them had started their sexual activity. Males present themselves as more sexually active, starting earlier, and have more sexual partners than females. Participants reported an adequate knowledge perception about STDs and contraception methods, however we found different patterns on specific STDs and contraceptive methods, according to gender, age, and sexually active status. Our results help design specific interventions to reach youth, community, and healthcare providers, pointing out the value of bringing people to the center of health policies.

## 1. Introduction

Adolescence is a stage of rapid growth that usually comprises the age from 10 to 18 years old [1]. It encompasses physiological, psychological, and social changes, crucial to individual identity and autonomy development, comprising emotional and sexual experimentation [2]. The development of secondary sexual characteristics and the reproductive system maturation are important dimensions of adolescence, often associated with arousing of sexual activity. The demand for autonomy and self-affirmation may increase risk-taking behavior in adolescents, making them particularly vulnerable to Sexually Transmitted Diseases, STDs [3]. Young adults, typically persons aged between 18 and 24, are also an important target concerning reducing health-risk behaviors’ adoption [4]. Therefore, it seems to be adequate to consider both age groups, adolescents, and young adults, as one, targeting a preventive approach.

STDs represent an important worldwide public health problem, both in industrialized and developing countries. The World Health Organization [5] estimates that more than 1 million people acquire a sexually transmitted disease every day, half of them between people aged between 15 and 24, as stated by the Centers for Disease Control and Prevention, CDC, although just one quarter relates sexual intercourse [6]. The reasons for this increased risk are early sexual behavior, multiple sexual partners, failure to use barrier protection, and inadequate access to healthcare services [7].

Scientific information regarding contraception and STD and risk behavior is wide available. The greater challenge for health policies is to provide an effective approach for adolescents and young adults, able to promote healthy sexuality, while prevent adverse outcomes as diseases, unintended pregnancy, or sexual violence. High-risk sexual behaviors include having two or more sex partners in the past 3 months, using alcohol or drugs at sex, unprotected sexual intercourse, without condoms or contraceptives, pregnancy during adolescence, being forced to have sex, and physical dating violence [4]. Sexual initiation is associated with sociodemographic and lifestyle characteristics, including adolescents’ involvement with alcohol, tobacco, and cannabis [8]. Early sexual initiation is commonly defined as the first sexual intercourse at underage of 16 and is associated with adverse sexual health outcomes [8], suggesting sexual debut with insufficient preparation. It is crucial to increase sexual competences at first intercourse, comprising using properly contraception, autonomy of decision, both partners willing it, and that it is happening at the “right time” [9]. Therefore, providing a proper preventive medicine approach before first sexual intercourse could be important, based on the integration of sexual education classes in school, the creation of effective channels providing information from healthcare professionals, and the improvement of communication regarding sexuality with their parents [10]. Reproductive health school-based interventions are widely accepted but there is little evidence that alone have a significant impact in reducing STDs or adolescent pregnancy [11]. Promoting healthier sexual behaviors should comprise interventions in schools and community, supporting access to environments of healthy relationships and responsible decision making about sexual activity [12]. Research regarding STD knowledge targeting Berliner adolescents shows that they present unsatisfactory levels of knowledge ahead of HIV, significant differences among sociodemographic characteristics [13]. An investigation regarding Italian University first year students showed that they present poor knowledge of STIs and their prevention [14]. A North American study suggests that STD literacy among late adolescents is correlated with gender, nationality, sexual identity, testing history of these infections, knowing someone with STDs, nationality, prior school-based programs, and receiving information from healthcare professionals [15]. 

Most studies focus on school-based populations and exclude adolescents who left school, either because they had finished their academic courses or because they had dropped out of school. Studying young people outside the school is required for an enhanced global perspective, supporting upgraded health promotion strategies [16].

Improving communication may lead suitable strategies regarding a healthier sexuality life. Knowledge about adolescents and young adult’s information concerning STDs, contraception, and their main counselors, can be an important step into this goal. This study aims to characterize adolescents and young adults’ knowledge and attitudes about sexually transmitted diseases and contraception, prospecting for their main trusted counseling sources, to heighten the impact of education for health promotion.

## 2. Methods

### 2.1. Type of Study

We conducted a cross-sectional, population-based, self-report survey of individuals aged between 14 and 24 years from Paredes, a county in northern Portugal. Paredes is a semi-urban region 35 km away from Porto (region’s capital), with good transportation accessibility. This county has a significant industrial activity, namely, furniture industry. In recent years, tertiary sector has grown, employing now 51.3% of the active workforce, according to official data, Census. Paredes has about 87 thousand inhabitants across a 157 km^2^ area, 18.6% of whom are under 15 years old. Paredes is one of the ten youngest Portuguese counties.

### 2.2. Sampling

Considering 2011 Portuguese National Census data, Paredes has 12.312 inhabitants aged between 14 and 24, living in 24 different civil parishes. We used stratified sampling, considering the proportion of inhabitants in civil parishes, assuming a design effect of 2.00. We estimated a sample size of 746 participants, for a 95% confidence interval, CI, and a maximum error of 5%. A detailed description of the sampling methodology is published elsewhere [16].

We used the random route method [17] to select participants for inclusion. We followed a strict protocol for choosing random streets, house numbers, dwellings, and the residents at each home who met the inclusion criteria, selecting a participant per home. If at a selected home there was more than one person eligible and available to participate, we chose the one whose birthday was closest to the interview date. The refusal to answer was respected by proceeding to the next street, under the protocol, until obtaining the required number of participants according to the calculated sample size.

### 2.3. Participants

All Paredes inhabitants aged 14 to 24 years were eligible for participation. Participants, their parents, or guardians (if they were under 16 years old), subscribed written informed consent. We excluded persons who had been away from home for more than 6 months, temporary visitors, guests, those who did not speak Portuguese, and those without physical or mental ability to answer the questionnaire.

### 2.4. Questionnaire

The questionnaire included six dimensions: social characteristics; education, occupation, and expectations; nutrition, physical activity, body image; social identity, daily routines, youth behavior; health issues; and demographic characteristics. For this analysis, we used the following variables: demographic characteristics (address, age, biological sex, civil status, occupation, economic sector of active workers), youth behaviors (alcohol misuse, tobacco or other psychoactive substances experimentation), health issues (psychological symptoms presence - having sleep disturbs or feeling sad, nervous, or angry, at least once a week), currently sexually active status, main counselors concerning health topics, awareness and knowledge about STDs (general knowledge perception; specific knowledge perception about the diseases - chlamydia, gonorrhea, hepatitis B, hepatitis C, syphilis, genital herpes, HIV, HPV and trichomonas) and awareness and knowledge about contraception (general knowledge perception; specific Knowledge perception about contraceptive methods, natural methods, birth control pills, condom, vaginal ring, contraceptive implant, oral emergency contraception, abortion).

Knowledge perception was evaluated using a Likert scale, from 1 (less knowledge) to 5 points (greater knowledge), also including the option “never heard about it” (specific knowledge perception). Variable was categorized as adequate if participants rated it from 4 points to 5 points, and inadequate if participants rated it from 1 point to 3 points, or as “never heard of”. Participants rated their general knowledge about STDs and contraception as excellent, good, sufficient, insufficient, very insufficient, or never heard, also including the answer “never heard of”. The variable was categorized as adequate if participants rated it as excellent” or “good”, and as inadequate if rated “sufficient”, “insufficient”, “very insufficient” and “never heard of”.

The questionnaire was pilot tested with a group of teenagers who were not part of the study population to check its validity and comprehensibility. 

### 2.5. Data Collection Methods

Data were collected from January to September 2014, using a team of interviewers. They were local volunteers (Paredes Town Hall Voluntary Program), trained to deliver the questionnaire using the following protocol: participants’ selection under the procedure described above, manual delivery of the form and informed consent, collection and sealing of questionnaires and forms using different envelopes (ensuring anonymity). All interviewers were supported and supervised by study coordinators.

All questionnaires with more than half of the items filled were considered for analysis. Incomplete responses were treated as missing data. After excluding non-valid questionnaires, additional 17 participants were selected under a protocol to reach the required sample size. 

### 2.6. Ethical Issues

The ethics committee of São João Central Hospital in Porto and the University of Porto Faculty of Medicine approved this study (decision 197/2013, 28/06/2013). The study procedures were in accordance with the Declaration of Helsinki and the Oviedo Convention on the rights of individuals taking part in biomedical investigations. 

All participants (or their parents or guardians) signed an informed consent form. The questionnaires were completely anonymous and were sealed in envelopes after being collected.

## 3. Statistical Analysis

We used descriptive and inferential statistics to calculate the prevalence. For hypothesis testing, we used Student’s *t*-tests, or nonparametric tests, and Chi-squared, or Fisher test, depending on the variables. The Kolmogorov–Smirnov test was used to check for normal distributions. The significance level was set at 0.05. Data were encoded and registered in a Microsoft Office Excel 2010 database and analyzed using IBM SPSS Statistics, version 24.0 (IBM Corp., Armonk, NY, USA). 

## 4. Results

The total sample included 746 participants, fifteen of them did not specify biological sex and were excluded. The analysis included 731 participants (54.6% females) with a mean age of 18.3 ± 2.8 years old. Most of the participants were still attending school (*n* = 516, 70.6%). Most of inquired were single (*n* = 689, 94.3%). Male gender reported more general risky behaviors, and female gender more psychological symptoms. Table 1 shows the sociodemographic characteristics by gender.

Describing participants’ sexual activity status, 50.5% (*n* = 347) had started their sexual activity by the time of the study, more males (57.0%; 95%CI: 51.5–62.6%) than females (45.3%; 95%CI: 40.3–50.3%), *p* = 0.003. After adjustment for age and gender, the prevalence of those who started the sexual activity was 54.3% (95%CI: 53.5–55.2%), 58.4% in males and 52.7% in females. The first sexual intercourse occurred on average at age of 16.7± 2.0 years, lower in males (16.2 ± 1.9 years) than in females (17.2 ± 2.1 years), *p* < 0.001. The prevalence of early first sexual intercourse, under 16 years old, was 22.2% (95%CI: 18.0–26.8%), higher in males, 28.9% (95%CI: 22.5–36.6%), than females, 15.2% (95%CI: 10.3–21.2%), *p* = 0.002. Sexually active participants had an average age of 20.20 ± 2.5, and not sexually active participants had an average age of 16.56 ± 2.1, *p* < 0.001. Participants’ mean number of sexual partners in the past year was 2.2 ± 0.6, it was significantly higher in males than in females (2.4 ± 0.5 vs. 2.0 ± 0.0, *p* < 0.001). The prevalence of participants who reported any STDs in the past year was 1,1% (*n* = 8), with no differences between genders.

The majority (82.0%, *n* = 574) considered their global knowledge about STDs as adequate: excellent (*n* = 255, 36.4%) or good (*n* = 319, 45.6%). Just 14.4% (*n* = 101) answered enough and 3.6% (*n* = 25) less than it.

We also studied knowledge perception about several STDs, as chlamydia, gonorrhea, hepatitis B, hepatitis C, syphilis, genital herpes, human immunodeficiency virus (HIV), human papilloma virus (HPV), and trichomonas. Figure 1 displays these results, prevalence of participants’ adequate STD knowledge. HIV, HPV, and genital herpes gave higher scores. The knowledge perception of participants about specific diseases presented a significant correlation with overall STDs perception, with higher correlation for hepatitis B in males (ρ of Pearson = 0.309, *p* < 0.001), and for HIV infection/AIDS in females (ρ of Pearson = 0.294, *p* < 0.001).

Table 2 shows knowledge about STDs by gender, age, and sexual activity. We grouped knowledge as adequate if it rated 5 or 4. Female gender has a significant higher proportion of adequate knowledge of hepatitis B and HPV. Participants over 17 years old have significant higher knowledge perception concerning STDs, except for HIV and general knowledge. Sexually active participants had a higher proportion of adequate knowledge of gonorrhea, hepatitis B and C, syphilis, and genital herpes. We did not find any significant difference between being and not being a student concerning the frequency of adequate disease knowledge.

Figure 2 displays the proportion of participants with adequate knowledge perception about several contraception methods. Most of adolescents and young adults (88.1%, *n* = 623) consider their knowledge as adequate: excellent (44.7%, *n* = 316), or good (43.4%, *n* = 307). Condoms and birth control pill methods had higher scores. Emergency contraception knowledge perception had higher correlation with general knowledge perception (ρ of Pearson = 0.396, *p* < 0.001).

We analyzed the association of participants’ general knowledge STDs perception and general knowledge contraception perception, that is significant (ρ of Pearson = 0.795, *p* < 0.001).

Table 3 shows participants proportion of adequate knowledge regarding contraception by gender, age, and sexually active status. We did not find any significant association between knowledge perception and being or not a student. Female gender had significant higher knowledge perception concerning contraceptive implant. Older participants considered having higher knowledge about contraception, except for contraceptive implant and vaginal ring. Being sexually active was associated with higher knowledge scores of general contraception, birth control pills, condom, and oral emergency contraception.

We asked the participants to identify persons with whom they felt more comfortable talking about sexuality and asking for advice. They identified friends (*n* = 167, 31.45%), mother (*n* = 147, 27.68%), and partner (*n* = 108, 20.34%) as trusted counselors. Healthcare providers and teachers were minor sources of information, accounting for only 5.46% (*n* = 29) and 2.07% (*n* = 11), respectively. Among whom that use to talk about sexuality with healthcare providers, we found that the majority were female (*n* = 44, 72.13%), over 17 years old (*n* = 40, 65.57%), and sexually active (*n* = 36, 62,07%). Table 4 shows health counsellors by sex, age, and sexually active status.

We did not find a significant association between knowledge level and main counselor concerning being sexuality active.

## 5. Discussion

Our results show that males are more sexually active, start younger than females, have more early sexual intercourse and more partners. This profile configures a set of risks and opens opportunities for intervention. As far we know this is the first Portuguese study using a population-based sampling. Even in international literature, most of studies include specific groups like school-based, or institutional, representing potential selection bias. 

Most participants feel that their knowledge is adequate both STDs and contraception in a global analysis and specifically about each issue. Differences in sociodemographic factors, as gender, age, and sexually active status, allow us to personalize our approach to improve its efficacy. Also, we found that family and friends are the main counsellors on sexuality issues, although fathers seem to be more relevant to males, mothers to females and partner to sexually active and older.

In our sample, 50.5% of participants were sexually active, with a mean age of first sexual intercourse of 16.7 years old, 16.2 years old in males. Miranda (2018) found a proportion 58% sexually active adolescents in a population of Portuguese high school and college students. The first intercourse occurred at 16.4 years, lower in males [18]. Mendes and Ferreira studies had participants with different age range and time of enrolling that may explain different results, compared to the present study. Mendes noticed a mean age of first sexual intercourse of 15.6 years, considering a Portuguese 13-to 19-year population [19], and Ferreira of 15.5 years, with girls being slightly older, considering the population of Porto County students at high school in 2005 [20]. Our findings show that sexual education and literacy should be accessible at least to adolescents under 14 years, to reach adolescents before their sexual debut, 16.7 ± 2.0 years. Preventive strategies should reach particularly male gender since males present early sexual intercourse and more sexual partners.

The prevalence of the first sexual intercourse under 16 years was 22.2% among those with a sexually active life. Irish research on students of 15 - to 18-years concluded that 68.3% of those with sexuality active life have an onset of sexual intercourse under 16 years, with equivalent proportions for both genders [8]. Kann reported sexual initiation before 14 years old in 39.3% of New York high school students, associated with sexual risky behaviors [4]. Rada, in Romania, estimated an overall average age of sexual initiation at 18.5 years, with 7.2% under 16 years, in the general population between 18 and 74 years [10]. We did not ask participants about sexual risky practice; however, we may consider the early first intercourse as a possible measure of adolescents becoming sexually active before developing physical and mental maturity. The prevalence of the first sexual intercourse under 16 years supports the importance of reaching young adolescents into sexual education.

In our study the mean number of sexual partners was 2.2 in past year (slightly higher in boys). Gravata found 1.9 partners during the last 6 months among foreign exchange students in Portugal from 2012 to 2015 [21]. In our sample, there were only eight STDs reported. Previous Portuguese studies reported a higher incidence of STDs, between 2.1% [20] and 7.8% [21]. Young people undervalue their illness, Haller in Australia [22] noticed a gap between young people’s perception of illness and their presentations to family doctors. Alternatively, healthcare services are less accessible to young people than other groups [23]. A systematic review concerning reproductive health show that youth-friendly family planning services have better outcomes [24]. So, we can consider that undervaluation of illness, or healthcare service misuse by young people may lead on low STD perception and low report. Fitting health services to adolescents and young adults’ expectations, may improve their accessibility, management of STDs, and health literacy.

Adolescents and young adults tend to consider adequate their knowledge about STDs, with higher results regarding HIV, HPV, and genital herpes. These diseases have a high public exposure HIV is a major public health problem, with high media coverage. Alternatively, the pre-exposure prophylaxis of HIV is available on public health services, that could upgrade the awareness of this STD. HPV is also a relevant issue in population since it is a target for immunization in the National Vaccination Program, and the relation with cervical cancer. The recent extension of HPV vaccine to males will certainly reduce the difference between genders, still clear in our results. Although genital herpes is one of the most common STDs worldwide [25], we think the perception of knowledge is somehow due to confusion with labial herpes, much more common in younger and adolescents [26]. Participants’ knowledge perception about chlamydia, trichomonas, and gonorrhea was lower that of other. The study of Fronteira [27], involving 1557 teenagers, showed high unfamiliarity of Portuguese high-school students with chlamydia, with just 12% who had ever heard of it, lower proportion than reported by Estonian, 51%, Belgian, 31%, or Czech adolescents, 29%. Being sexually active is associated with a higher score regarding knowledge perception of gonorrhea, syphilis, hepatitis C, and genital herpes, which was also noted in other studies [28], we also found that people sexually active are often more aware of DSTs, as do elderly and females [29]. The perception of better knowledge may be understood as more worry and more awareness of these issues. However, the pattern suggests that information comes mainly across the media exposure, conditioned by the often-hyped news of the day, and potentially biased by unclear lobbying, thus transmitting less-quality information. It is crucial to provide good sources of information and a system that certifies its reliability. Prophylaxis procedures as vaccination or pre-exposure medication may constitute an opportunity to promote health literacy regarding STDs by health professionals. Preventive strategies should highlight other STDs beyond HIV.

On attending contraception, most participants considered their knowledge adequate. Condoms and birth control pills were the most pointed methods, corresponding to the most used in these years [30]. Although other methods are equally available for free distribution in Primary Health Care settings, vaginal ring and contraceptive implant are not so recognized in our population, seeming to be a deficit of true autonomy in the access to contraception since the difference of information. A European survey reported that lower use of long acting of reversible contraceptives is associated with lower knowledge about this contraceptive option and perception of healthcare professionals provide less information about these contraceptives [31]. Females are more awareness of implants, and they also consider the healthcare provider as the main counselor. There is a way to be done. It is not enough to produce lots of guidelines about health conditions, based on the best evidence. It is crucial to include patient’s values and expectations [32]. A family planning program of teenagers must be really adjusted to this population, with specific accessibility, proper agendas, and even different places, able to maximize the relevance of intervention, both medical and educational, to improve their health, and to avoid the use of emergency contraception and abortion, two aspects very present in this population. We must improve awareness of long-acting reversible contraceptives among this population. Unfortunately, we must emphasize that education on contraceptive methods and on safe sex is not a guarantee that adolescents will use the right methods and that they will use it properly, but we have no doubt that information is fundamental for the decision making. 

The quality of knowledge and the level of competence achieved depend on the trustfulness of the information sources. In this study, parents, friends, and partners were identified as the main advisors to sexual issues. Study about Italian University first year students show that they discuss more often STDs with General Practitioner than partners, not according to the present study [14]. Parents are crucial in this process, with fathers appearing specially for males and mothers for both sexes, raising the importance of educating the direct family, creating a healthy environment and for them to be active educators themselves [16]. Our results support the potential of strategies targeting both adolescents and their families.

This role is gradually transferred to the partner as they become sexually active and older. Females often use the healthcare providers for information. The HPV vaccine and the menarche are certainly between the determinants for better acceptance and access. Teachers are chosen preferably by younger and by those who have uninitiated their active sexual life, revealing their importance in primary prevention, to transmit general information. Therefore, we see that sources change over time as the type of closeness, accessibility, trust, or empathy, making it important to understand exactly what the situation of each adolescent is to choose the better approach. Nevertheless, teachers and physicians should play a more visible role. Sexual education is mandatory in the Portuguese education system and has been included in the curricula of the basic and secondary levels since 2009, although, in practice, only about half of the students actually have access to it [19], or even less in other countries [33]. Structured and unbiased information is crucial to create competence and must be an aim for educators and doctors, even though we know that the access to good information is not a guarantee that adolescents will follow in the practice [34]. Portuguese Primary Care offers a family planning support, including the free distribution of contraceptives, but with low usage by younger [20,27], as compared to older women [30], more frequenters, and with greater confidence. This is a huge problem. Adolescents lack accessibility to Health Centers, and it may compromise the wanted outcomes. It urges to bring them back, probably by launching health providers out of the building, taking advantage of the school classroom environment, for instance, not only to inform but to improve this access, when needed, and seizing the opportunities to warm welcome younger into healthcare, promoting the accessibility. Moreover, the national programs must be targeted to the population, involving people, and understanding them, in their values, opinions, expectations and fears, and this implies some change in the current guidelines [32]. Media campaigns, internet and social networks are also available tools and potentially useful [10], with high acceptance and are easily accessible by the population. However, much information presents several biases and may compromise the literacy instead of improving it. Health services and schools should evaluate their practices and potential regarding sexual health promotion in to enhance youth-centered approaches.

## 6. Limitations and Strengths

Adolescents and young adults’ habits studied may be influenced by memory bias, considering that some of them do not refer to recent events, as experimentation, or sexual debut. Participants’ knowledge perception is not synonymous of factual and useful knowledge, further investigations evaluating knowledge and its impact should be useful. The wide range of participants age and the evidence of influence of age in habits and perceptions may suggest the relevance of more studies considering separate groups according to this attribute, despite common characteristics. This study did not assess specific knowledge or adherence to prophylaxis, such as pre-exposure strategies or vaccination, that could provide useful information. Although the data were collected in 2014, there is no evidence that pos-COVID-19 time significantly changed our findings. We chose a robust participants’ enrolling method, random, presential and embracing one, including those adolescents and young adults out of school trail. Although the questionnaire has not been validated, it was updated through a pilot test, by other hand it comprised demographic, youth behavior, expectations, and perception data, acknowledging a wider perspective.

## 7. Conclusions

Portuguese adolescents and young adults are mostly students, and about half are sexually active. Males present themselves as more sexually active, starting earlier and having more sexual partners than females. Although generally, adolescents and young adults report adequate knowledge perception about STDs and contraception methods, we found different patterns on specific STDs and on contraceptive methods, according to gender, age, and sexually active status. Sexual education is crucial for a pleasant, comfortable, and responsible sexuality. Our results help design specific interventions to reach each player, between younger, and between every other one around them, including, of course, the healthcare providers and the need to really bring people to the center of the system, changing the system itself, if necessary.

## Figures and Tables

**Figure 1 ijerph-19-13933-f001:**
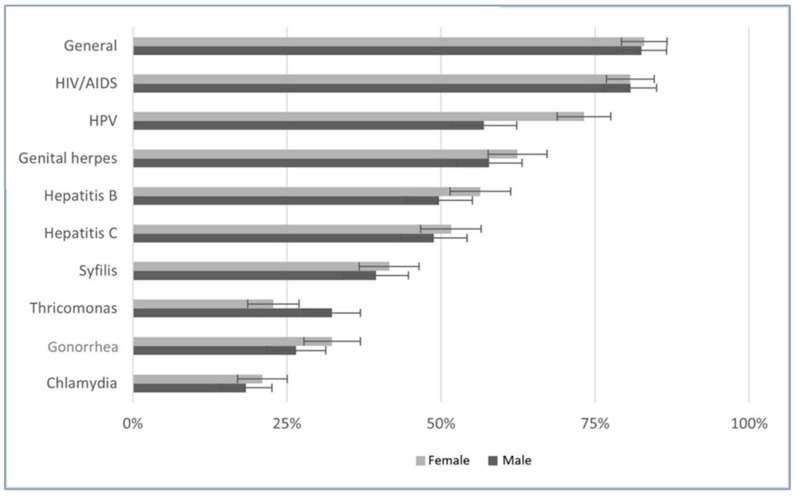
Adequate Knowledge about STDs.

**Figure 2 ijerph-19-13933-f002:**
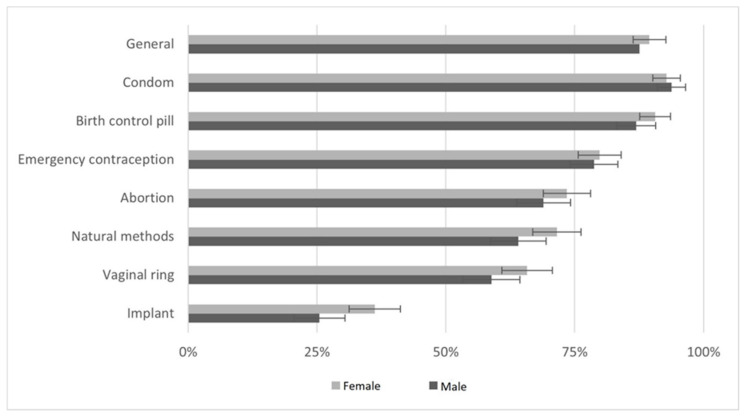
Adequate knowledge about contraception.

**Table 1 ijerph-19-13933-t001:** Sociodemographic characteristics by gender.

Characteristics		Gender	
		Male (*n* = 332)	Female (*n* = 399)
Age	Years, mean ± SD	18.5 ± 3.0	18.3 ± 2.8
	Under 18 years, *n* (%)	143 (43.0)	174 (43.6)
Civil status	Single, *n* (%)	312 (94.0)	377 (94.4)
	Married, *n* (%)	9 (2.7)	11 (2.8)
	Other, *n* (%)	3 (3.3)	11 (2.8)
	No response, *n* (%)	9 (2.7)	-
Occupation	Active worker, *n* (%)	77 (23.3)	62 (15.5)
	Unemployed, *n* (%)	40 (12.0)	53 (13.3)
	Student, *n* (%)	225 (67.8)	291 (72.9)
Economic sector (active workers)	Primary, *n* (%)	-	-
	Secondary, *n* (%)	32 (41.6)	20 (32.3)
	Tertiary, *n* (%)	40 (51.9)	33 (53.2)
	No response, *n* (%)	5 (6.5)	9 (14.5)
Risky behaviors and psychologic well-being	Have been drunk, *n* (%)	112 (40.9)	84 (27.7)
	Have experimented tobacco, *n* (%)	194 (61.2)	187 (49.0)
	Have experimented psychoactive substances, except alcohol and tobacco, *n* (%)	71 (21.8)	38 (9.7)
	Psychological symptoms present, *n* (%)	153 (47.5)	238 (60.6)

**Table 2 ijerph-19-13933-t002:** Global, and particular knowledge about STDs by gender, age, and sexually active status.

	STDs (General)		Chlamydia		Gonorrhea		Hepatitis B		Hepatitis C		Syphilis		Genital Herpes		HIV		HPV		Trichomonas	
Gender																				
Male (*n* = 332)	281/322 (87.3)	0.522	56/327 (17.1)	0.204	84/328 (25.6)	0.067	160/327 (48.9)	0.045	158/328 (48.2)	0.371	126/327 (38.5)	0.524	186/326 (57.1)	0.151	264/328 (80.5)	0.968	185/328 (56.4)	<0.001	71/326 (22.1)	0.866
Female (*n* = 399)	342/385 (88.8)		83/398 (20.9)		126/396 (31.8)		225/399 (56.4)		205/398 (51.5)		161/394 (40.9)		248/398 (62.3)		320/397 (80.6)		292/399 (73.2)		90/398 (22.6)	
Age																				
<18 years (*n* = 317)	261/306 (85.3)	0.027	40/315 (12.7)	<0.001	67/314 (21.3)	<0.001	140/315 (44.4)	<0.001	128/315 (40.6)	<0.001	88/313 (28.1)	<0.001	165/313 (52.7)	0.001	244/314 (77.7)	0.089	193/315 (61.3)	0.041	54/314 (17.2)	0.004
≥18 years (*n* = 409)	361/398 (90.7)		96/405 (23.7)		140/405 (34.6)		240/406 (59.1)		232/406 (57.1)		195/403 (48.4)		265/406 (65.3)		336/406 (82.8)		279/407 (68.6)		106/405 (26.2)	
Sexually active																				
No (*n* = 384)	280/329 (85.1)	0.027	55/335 (16.4)	0.098	78/336 (23.2)	0.002	161/336 (47.9)	0.008	152/336 (45.2)	0.014	105/332 (31.6)	< 0.001	185/335 (55.2)	0.019	264/335 (78.8)	0.377	215/337 (63.8)	0.416	65/337 (19.3)	0.084
Yes (*n* = 347)	303/334 (90.7)		74/346 (21.4)		117/344 (34.0)		201/346 (58.1)		189/346 (54.6)		161/345 (46.7)		221/345 (64.1)		282/346 (81.5)		231/346 (66.8)		85/343 (24.8)	

**Table 3 ijerph-19-13933-t003:** General and particular knowledge about contraception by sex, age, and being sexually active.

	Contraception (General)		Natural Methods		Birth Control Pills		Condom		Vaginal Ring		Contraceptive Implant		Oral Emergency Contraception		Abortion	
Gender																
Male (*n* = 332)	281/322 (87.3)	0.522	205/318 (64.5)	0.082	282/326 (86.5)	0.184	304/327 (93.0)	0.522	189/326 (58.0)	0.078	84/327 (25.7)	0.006	256/327 (78.3)	0.716	224/325 (68.9)	0.372
Female (*n* = 399)	342/385 (88.8)		274/388 (70.6)		357/398 (89.7)		364/397 (91.7)		255/396 (64.4)		139/395 (35.2)		316/398 (79.4)		280/389 (72.0)	
Age																
<18 years (*n* = 317)	261/306 (85.3)	0.027	194/306 (63.4)	0.034	262/315 (83.2)	< 0.001	2787314 (88.5)	0.001	184/315 (58.4)	0.163	87/314 (27.7)	0.152	228/314 (72.6)	< 0.001	206/308 (66.9)	0.048
≥18 years (*n* = 409)	361/398 (90.7)		281/396 (71)		374/405 (92.3)		386/406 (95.1)		256/403 (63.5)		132/404 (32.7)		340/407 (83.5)		297/403 (73.7)	
Sexually active																
No (*n* = 384)	280/329 (85.1)	0.027	216/329 (65.7)	0.247	286/337 (84.9)	0.003	300/336 (89.3)	0.005	202/337 (59.9)	0.434	103/337 (30.6)	0.839	247/338 (73.1)	< 0.001	231/331 (69.8)	0.435
Yes (*n* = 347)	303/334 (90.7)		234/664 (69.9)		317/244 (92.2)		328/345 (95.1)		215/342 (62.9)		107/342 (31.3)		296/345 (85.8)		248/342 (72.5)	

**Table 4 ijerph-19-13933-t004:** People that participants feel more comfortable talking about sexuality.

	Father		Mother		Sibling		Other Relatives		Partner		Friends		Professors		Health Care Providers	
Gender																
Male (*n* = 304)	58 (19.1)	< 0.001	108 (35.5)	0.029	26 (8.6)	0.040	19 (6.3)	0.890	78 (25.7)	0.360	123 (40.5)	0.283	13 (4.3)	0.672	17 (5.6)	0.007
Female (*n* = 384)	29 (7.6)		168 (43.8)		18 (4.7)		25 (6.5)		87 (22.7)		140 (36.5)		14 (3.6)		44 (11.5)	
Age																
<18 years (*n* = 295)	51 (17,3)	0.002	142 (48,1)	<0.001	19 (6,4)	0.892	26 (8,8)	0.028	33 (11,1)	<0.001	112 (38,0)	0.853	23 (7,8)	<0.001	21 (7,1)	0.148
≥18 years (*n* = 388)	36 (9,3)		130 (33,5)		24 (6,2)		18 (4,6)		132 (34,0)		150 (38,7)		4 (1,0)		40 (10,3)	
Sexually active																
No (*n* = 317)	46 (14,5)	0.107	153 (48,3)	<0.001	19 (6,0)	0.906	18 (5,7)	0.829	29 (9,1)	<0.001	127 (40,1)	0.272	19 (6,0)	0.006	22 (6,9)	0.075
Yes (*n* = 329)	34 (10,3)		103 (31,3)		19 (5,8)		20 (6,1)		126 (38,3)		118 (35,9)		6 (18,2)		36 (10,9)	

## Data Availability

Data is available under request directly to the main researcher.

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
