# Peer review of "Young People Awareness of Sexually Transmitted Diseases and Contraception: A Portuguese Population-Based Cross-Sectional Study"

_ijerph, 2022, doi:10.3390/ijerph192113933_

Round 1

Reviewer 1 Report

The paper presents the results of a study about STD and contraception awareness based on a questionnaire distributed among people aged 14-24 living in a small city in the north of Portugal in the year 2014. Participants were segmented by sex, age, occupation and other relevant dimensions. Questions regarded their sexual life and their knowledge about STDs and contraceptive methods.

I found this an interesting paper which is surely well organized in general. I only have two comments to make, one regarding the data used ands the other regarding the structure of the Discussion section.

Regarding the data used in the study, they date back to 8 years ago (2014). This is a long time nowadays, also considering that the Covid-19 pandemics has changed the way we interact in many social aspects. Young people today might have different behaviors and knowledge than people who were young almost a decade ago. Therefore I suggest that the issue is briefly discussed in order to underline why the study is still relevant.

I suggest some improvements to the Discussion section which would from one hand make it easier to read and from the other hand would better highlight its contribution. I say this because results in the Discussion section are presented in a somehow confused structure. Statistics, quotation from other sources, deductions and conclusions are mixed. The section starts with a paragraph about males. It then continues talking about the questionnaire’s answers regarding awareness. It then quotes other sources about mean age of first sexual intercourse. And so on, with each paragraph seeming to have no connection with the previous or the following one… Sometime results and suggestions are proposed, but they are hidden and diluted in the general, confused text: e.g. p. 9 “these results may support the strategies targeting promoting sexual competence should be accessible at least to adolescents under 16 years old, to reach adolescents be-fore their sexual debut.”

I think that a clear list of the most interesting results and of the actions they suggest would be more useful and easy to read. 

Minor issues: p. 7 – figure 2 caption says ‘figure 1’; p.  7/8  - several references to figures/tables appears in the text as ‘Error! Reference source not found’

Author Response

Please, check the attached file.

Reviewer 2 Report

The study is sound and the findings are well described and presented.  If I have a concern or criticism, from a sexual health intervention perspective, it is in the assumptions made by the authors or your lack of discussion about important issues happening in sexual health discourses.  For example, not going behind the male/female gender classification, not including knowledge about medical intervention like Prep etc.  You may have good justifications for your decisions, but these need to be discussed, either in your discussion, or possible limitation of the study.

Also, your introduction of the literature needs to better capture recent advances in sexual health interventions, and arguing why this study is needed, given that we have a lot of research on the general topic of young people and STI knowledge.

The discussion needs a stronger argument explaining what new insights have been found by this study.  

Author Response

Please, check the attached file.

Author Response

Please, check the attached file.
